Lionfish (Pterois spp.) invade the upper-bathyal zone in the western Atlantic

Gress Erika 1
Andradi-Brown Dominic A. dominic.andradi-brown@zoo.ox.ac.uk dandradibrown@gmail.com 2 3
Woodall Lucy 1 2
Schofield Pamela J. 4
Stanley Karl 5
Rogers Alex D. 1 2
1 Nekton Foundation, Begbroke Science Park , Begbroke , Oxfordshire , United Kingdom
2 Department of Zoology, University of Oxford , Oxford , Oxfordshire , United Kingdom
3 Operation Wallacea , Old Bolingbroke , Lincolnshire , United Kingdom
4 U.S. Geological Survey , Gainesville , FL , United States of America
5 Roatan Institute of Deepsea Exploration , West End , Roatan , Honduras
Reimer James
Electronic publication date: 2017 Aug 17
Publication date: 2017
Volume: 5
Electronic Location ID: e3683
Received 2017 May 29; Accepted 2017 Jul 23
Copyright year: 2017
License: This is an open access article, free of all copyright, made available under the Creative Commons Public Domain Dedication. This work may be freely reproduced, distributed, transmitted, modified, built upon, or otherwise used by anyone for any lawful purpose.
License URL: https://creativecommons.org/publicdomain/zero/1.0/

Keywords: Lionfish, Pterois, Deep sea, Bermuda, Roatan Honduras, Invasive species, Upper bathyal, Depth record

Funding: Fisheries Society of the British Isles (FSBI) Ph.D. Studentship Student Conference on Conservation Science Miriam Rothschild Travel Bursary Programme XL Catlin and the Garfield Western Foundation US Geological Survey’s Invasive Species Program European Union’s Horizon 2020 research and innovation programme 678760 DAAB is funded by a Fisheries Society of the British Isles (FSBI) Ph.D. Studentship. EG was supported by the Student Conference on Conservation Science Miriam Rothschild Travel Bursary Programme for funding. This research was undertaken as part of the XL Catlin Deep Ocean Survey—Nekton’s Mission to the North West Atlantic and Bermuda. Nekton received support from the XL Catlin and the Garfield Western Foundation. This project was supported by the US Geological Survey’s Invasive Species Program. Any use of trade, product or firm names is for descriptive purposes only and does not imply endorsement by the US Government. This project has received funding from the European Union’s Horizon 2020 research and innovation programme under grant agreement No 678760 (ATLAS). This output reflects only the author’s view and the European Union cannot be held responsible for any use that may be made of the information contained therein. The funders had no role in study design, data collection and analysis, decision to publish, or preparation of the manuscript.

==============================
Non-native lionfish have been recorded throughout the western Atlantic on both shallow and mesophotic reefs, where they have been linked to declines in reef health. In this study we report the first lionfish observations from the deep sea (>200 m) in Bermuda and Roatan, Honduras, with lionfish observed to a maximum depth of 304 m off the Bermuda platform, and 250 m off West End, Roatan. Placed in the context of other deeper lionfish observations and records, our results imply that lionfish may be present in the 200–300 m depth range of the upper-bathyal zone across many locations in the western Atlantic, but currently are under-sampled compared to shallow habitats. We highlight the need for considering deep-sea lionfish populations in future invasive lionfish management.

Introduction

Non-native lionfish, first documented in the western Atlantic region in the 1980s (Schofield, 2009; Schofield, 2010), are considered a major threat to western Atlantic reef communities (Sutherland et al., 2010). Lionfish are benthic generalist predators, and their presence on shallow coral reefs has been associated with up to 65% decline in their prey fish biomass (Green et al., 2012), leading to overall declines in fish recruitment of up to 79% (Albins & Hixon, 2008). In some cases lionfish have been observed to feed on critically-endangered reef fish (Rocha et al., 2015). On both shallow reefs and mesophotic coral ecosystems (MCEs, reefs from 30 to approximately 150–180 m depth; (Hinderstein et al., 2010)), non-native lionfish are thought to cause increased algal cover by consuming herbivores and causing trophic cascades (Lesser & Slattery, 2011; Slattery & Lesser, 2014; Kindinger & Albins, 2017). Native to the Indian and Pacific oceans and Red Sea, lionfish in the western Atlantic have now been recorded from New York, USA in the north (Meister et al., 2005), to as far south as the southeastern coast of Brazil (Ferreira et al., 2015). In addition, there is a second lionfish invasion currently underway in the Mediterranean Sea (Kletou, Hall-Spencer & Kleitou, 2016). Two species of non-native lionfish have been recorded in the western Atlantic: Pterois volitans (Linnaeus, 1758) and P. miles (Bennett, 1828) (Hamner, Freshwater & Whitfield, 2007), though they are believed to be ecologically synonymous in their impacts to western Atlantic marine communities (Morris et al., 2009).

The majority of research on lionfish invasions has focused on shallow coral reefs (<30 m), mangroves and seagrass beds (Morris et al., 2009; Claydon, Calosso & Traiger, 2012). However, recent studies have highlighted their widespread presence on MCEs across the western Atlantic invaded range (Andradi-Brown et al., 2017a), which is unsurprising, as they have been recorded on MCEs in many locations in their native range. For example, P. miles at 65 m in the Red Sea (Brokovich et al., 2008), and P. volitans at 75 m in New Caledonia (Kulbicki et al., 2012), 61 m in Micronesia (Andradi-Brown et al., 2017a), 61 m in the Philippines (Andradi-Brown et al., 2017a), and at 80 m in American Samoa (Wright, 2005). With two exceptions (see next paragraph), MCEs represent the deepest depths lionfish have been previously reported from in the western Atlantic. For example, from remote operated vehicle (ROV) surveys: 112 m in the northwestern Gulf of Mexico (Nuttall et al., 2014), 100 m off North Carolina, USA (Meister et al., 2005), 126 m on the Desecheo Ridge west of Puerto Rico (Quattrini et al., 2017), and 167 m on the Conrad Seamount in the Anegada Passage (Quattrini et al., 2017). Lionfish have also been observed at 120 m from submersible dives in Honduras (Schofield, 2010), and collected from trawl surveys >80 m depth in the eastern Gulf of Mexico (Switzer et al., 2015). In addition, diver-based surveys on MCEs have reported sightings in the 30–100 m range in Puerto Rico (Bejarano, Appeldoorn & Nemeth, 2014), Bermuda (Pinheiro et al., 2016), and the Lesser Antilles (De León et al., 2013). Therefore, it has been suggested that lionfish have widely colonised MCEs across the western Atlantic (Andradi-Brown et al., 2017a).

In August 2010, while conducting submersible surveys off Lyford Cay, Nassau, The Bahamas, lionfish were observed at 300 m (pers. comm. from RG Gilmore in: Albins & Hixon, 2013; McGuire & Hill, 2014). While in Curaçao, the Curasub has reported observing lionfish regularly down to 247 m depth (Tornabene & Baldwin, 2017). To our knowledge these sightings represent the maximum known depth distribution of lionfish in the western Atlantic, and the only records of lionfish in the deep sea (defined as >200 m depth; Rogers, 2015). It is not clear whether these sightings represent isolated incidents of lionfish reaching these depths, or whether lionfish more regularly use habitats in the >200 m depth range, but they have not previously been recorded because of limited surveys within this depth range.

Figure 1 Map showing location of deepest observed lionfish for (A) Bermuda and (B) Roatan, Honduras.

Inset maps indicate the locations of Bermuda and Roatan respectively relative to the western Atlantic region. In (A) the dashed line indicates the 50 m depth contour to show the outline of the Bermuda platform. The reef drops off steeply at this location, such that our 304 m lionfish observation is close to the 50 m depth contour.

In this study we report visual observations of lionfish >200 m depth in two new locations within the western Atlantic region: Bermuda and Roatan, Honduras. We also consider other lionfish records that could potentially indicate that lionfish may be more widespread at >200 m depth across the western Atlantic range.

Methods

Bermuda is a series of islands located far off the continental shelf in the northwestern Sargasso Sea (Fig. 1A). The islands exist on a large shallow-water platform (approximately 20 m depth, 623 km2 area) which are the eroded remains of a Meso-Cenozoic volcanic peak (Coates et al., 2013). The platform is surrounded by a shallow slope, which transitions into near-vertical walls at around 100 m (Coates et al., 2013). While deep reef areas of Bermuda are poorly studied, with few observations below mesophotic depths, there are established MCE communities around Bermuda to at least 80 m (Pinheiro et al., 2016). MCE to deep-sea benthic organisms and benthic-associated fish surveys were undertaken during daylight hours using the Nemo and Nomad Triton 1000-2 class submersibles (Vero Beach, Florida, USA) down to 300 m depth around the edge of the Bermuda platform during July and August 2016 as part of the Nekton Foundation/XL-Catlin Deep-Ocean Survey –Mission 1 (www.nektonmission.org). In total, 17 dives were conducted to 300 m between both submersibles. Research permits for Bermuda were issued by the Department of Environment and Natural Resources, Bermuda (No. 2016070751).

In contrast, Roatan is an island in the Caribbean Sea located off the north coast of mainland Honduras (Fig. 1B). Roatan is approximately 50 km long and 2–4 km wide, and has a total land area of about 200 km2. This island is surrounded by shallow fringing coral reefs, which transition into MCEs at increased depths. The Roatan Institute of Deepsea Exploration conducts commercial submarine tourism, using the Idabel submarine allowing tourists to observe deep-sea habitats to 610 m depth. With year-round operations from Half Moon Bay, West End, Roatan, Idabel conducted 224 dives ≥300 m between Jan 2015–April 2017. During March 2017 visual observations of benthic communities and their associated fish communities were conducted on a night dive to 300 m depth. Visual/video lionfish observations in Roatan were covered under the Roatan Institute of Deepsea Exploration operating permit issued by the Municipalidad de Roatan (No. 1391).

To identify other records of deep lionfish we examined 6,814 lionfish records from the US Geological Survey Nonindigenous Aquatic Species database (USGS-NAS, 2017). Lionfish records in the database have been gathered from media reports, scientific publications and direct reports to the database managers. All records contain a GPS location, and in some cases a short description of the conditions under which the lionfish was observed and/or a photo of the lionfish. In some cases the descriptions accompanying records included depth information, though this is not formatted in a consistent way (for example using different units such as metres, feet, fathoms) and contained within a larger text record description. We initially viewed these descriptions to identify any records directly stating lionfish observations at depths ≥200 m, converting any depth information provided into metres for consideration. To further identify potential lionfish records from ≥200 m depth, we downloaded the 2014 General Bathymetric Chart of the Oceans (http://www.gebco.net) 30 arc-second interval grid bathymetry for the western Atlantic region. We used the raster package (Hijmans, 2015) in R (R Core Team, 2013) to identify approximate depths of all lionfish records based on GPS location. All records associated with bathymetry ≥200 m depth were individually reviewed and classified as potential deep-sea individuals, or excluded. Records were excluded for any of the following reasons: (i) specific depth information was available in the record indicating the fish was <200 m depth, (ii) the record reports that the observation was made by a diver or snorkeler, (iii) the location of the record is a well known/established shallow reef diving/snorkelling site, or (iv) the lionfish were collected by hook-and-line making it highly unlikely they were from ≥200 m depth. Raw data are available from the US Geological Survey Nonindigenous Aquatic Species database (USGS-NAS, 2017), and ESM1 contains a list of all lionfish GPS locations present in the database at the time of analysis that were used in this study. ESM2 contains the raw R code used to assign a depth to each lionfish record.

Results

In Bermuda during daytime dives on 28 July 2016 off the northeastern edge of the Bermuda platform at 32.483683 N, 64.59395 W (Fig. 1A; GPS coordinates in WGS84 format), multiple lionfish were observed. The deepest lionfish were a single individual observed at 304 m depth, and another individual at 297 m (Fig. 2). Water temperature was recorded on the submersible during the dive as 19.7 °C at 300 m. The laser points in Fig. 2B are 0.25 m apart, suggesting an approximate total length of 21 cm for this individual at 297 m.

Figure 2 Lionfish at 297 m depth off the northeastern slope of the Bermuda platform.

(A) The lionfish resting on the reef is indicated within the red circle. Other fish species shown are Gephyroberyx darwinii and cf. Pronotogrammus martinicensis. (B) Lionfish swimming over the benthos. The laser dots are separated by 0.25 m. Both (A) and (B) show the same individual that swam across the benthos as disturbed by the submersible.

In Roatan, on 11 March 2017 off Half Moon Bay, West End at 16.308565 N, 86.596681 W (Fig. 1B) five lionfish were observed and photographed down to a depth of 240 m (Fig. 3). Individuals were seen in on lower-MCEs (180 m; Fig. 3A), and the upper-bathyal (240 m; Fig. 3B). Water temperature was recorded on this dive as approximately 15 °C at 240 m. However, with year-round tourist submarine dives operating from Half Moon Bay visiting deep reef habitats ≥300 m (224 dives between Jan 2015–April 2017), the Idabel has regularly observed lionfish to a maximum depth of 250 m.

Figure 3 Lionfish off Half Moon Bay, West End, Roatan, Honduras.

(A) Lionfish swimming over the benthos at 180 m depth, and (B) two lionfish resting at 240 m depth.

When analysing records from the US Geological Survey Nonindigenous Aquatic Species database, no records were found explicitly stating a depth of observation ≥200 m. However, 186 records out of the 6,814 records were associated with bathymetry ≥200 m. Of these, after scrutinising the text descriptions, we excluded 185 records as being too shallow. Many of these records represented sites with steep walls spanning from shallow reefs to >200 m depth, and while the resolution of the available bathymetry suggested these were ≥200m, when checking the associated meta-data for these 185 records it clearly indicated that the lionfish were most-likely <200 m. The one record that we retained did not contain enough detail to confirm or reject it as a sighting from >200 m. Figure 4 shows the locations of this unconfirmed record, the previously confirmed 300 m lionfish observation in the Bahamas (Albins & Hixon, 2013; McGuire & Hill, 2014), the recorded observations in Curaçao at 247 m (Tornabene & Baldwin, 2017), and the locations of our deep-sea lionfish observations in Bermuda and Roatan. We have now added our new deep-sea lionfish observations to the US Geological Survey Nonindigenous Aquatic Species database.

Figure 4 Locations of confirmed and possible lionfish observations ≥200 m depth in the western Atlantic. The confirmed sightings represent our observations in Bermuda and Roatan, and the previously reported observations in the Bahamas (Albins & Hixon, 2013) and Curaçao (Tornabene & Baldwin, 2017). The unconfirmed sighting represent a record from the US Geological Survey Nonindigenous Aquatic Species database associated with bathymetry ≥200 m, though there is no direct information on the depth of the lionfish observation for this record.

Discussion

In this study we report deep-sea lionfish observations from the upper-bathyal zone in two new locations within the invaded western Atlantic lionfish range. Both in Bermuda, close to the northern limit of the lionfish overwintering invaded range (Eddy et al., 2016), and in Roatan, within the centre of the lionfish invaded range (Schofield, 2010), we report lionfish >200 m depth. Because of the large geographical distance between our observations, combined with the previous confirmed observations of lionfish >200 m from the Bahamas (pers. comm. from R.G. Gilmore in: Albins & Hixon, 2013) and Curaçao (Tornabene & Baldwin, 2017), we suggest that lionfish may be more widespread and common than presently understood in deep-sea habitats in the 200–300 m depth range and that this deeper aspect of the lionfish invasion has likely been under-sampled.

When searching the US Geological Survey Nonindigenous Aquatic Species database, we found one lionfish record located over bathymetry ≥200 m without stating a depth or giving any indication of depth. While the US Geological Survey Nonindigenous Aquatic Species records have been placed over bathymetry in previous studies, leading to the suggestion that lionfish may extend their maximum depth to 610 m (Johnston & Purkis, 2011), our results indicate depth records generated in this way must be treated with caution. The grid resolution of bathymetry available at a regional level is not sufficient to generate precise lionfish depth information over undersea structures such as walls and steep slopes, where large differences in depth occur within one raster grid square. For this reason, despite identifying 186 records associated with deeper bathymetry, 185 of these were excluded for containing either specific depth details or enough information to suggest that they were most likely shallower reef or MCE observations. Some of these excluded observations were from lionfish associated with oil and gas rigs, where lionfish were associated with the rig structure at shallower depths rather than actually with seabed benthic habitats. Therefore, from simply matching GPS locations with bathymetry, these records would appear to be >200 m and far from any shallower habitat, yet they actually represent shallower lionfish. As many of these records in the US Geological Survey Nonindigenous Aquatic Species database come from recreational and scientific divers and fisheries, we would expect these records to be biased towards shallow reefs where the majority of sampling has occurred. Therefore, it was not surprising that 6,628 of the 6,814 records were associated with shallow reef or MCE bathymetry. With this biased survey effort to the shallows, the lack of records >200 m in the database should be treated as an indication of under-sampling at depth, and not that lionfish are not present in the upper-bathyal.

It is not clear why differences in the maximum depth of observations exist between Roatan, our observations in Bermuda and previous observations in the Bahamas and Curaçao. While in Bermuda we observed lionfish to the maximum survey depth (304 m), in Roatan, despite 224 submarine dives to ≥300 m over the past 2.3 years, lionfish have not been observed deeper than 250 m. There are many possible explanations related to changing environmental conditions, such as temperature and light, or availability of prey. For example, lionfish are limited by temperature (Whitfield et al., 2014; Dabruzzi, Bennett & Fangue, 2017), with lab experiments suggesting they are unable to survive temperatures <10 °C, but crucially they ceased feeding at temperatures <16.1 °C (Kimball et al., 2004). While detailed temperature data across the depth gradient is not available for the locations we surveyed, water temperature was approximately 15 °C at 240 m in Roatan when we photographed lionfish in March 2017. Therefore, it is possible that the 250 m maximum depth of lionfish observations around Roatan may be caused by temperature limitation. In contrast, water temperature was 19.7 °C in Bermuda at 300 m, above the temperature of feeding cessation for lionfish (Kimball et al., 2004). This suggests that if temperature is the main limiting factor for maximum depth, we may expect lionfish to extend even deeper than 304 m in Bermuda.

Other factors such as light could also influence the maximum depth for lionfish. Lionfish are visual predators (Cure et al., 2012); therefore, despite previous studies indicating reef fish have high visual system plasticity to adapt to low light levels at depth (Brokovich et al., 2010), it is likely they will be limited by light. Bermuda has high light penetration (Fricke & Meischner, 1985; Coates et al., 2013), while Roatan suffers from higher sedimentation rates (Mehrtens et al., 2001; Harborne, Afzal & Andrews, 2001), likely reducing light penetration to lower levels than Bermuda. Further research is required to understand the ecological and physiological constraints on maximum lionfish depths.

Little is known about the potential impacts of invasive lionfish on the upper-bathyal zone. However, shallow reef research has suggested large declines in native reef fish abundance and recruitment are caused by lionfish (Albins & Hixon, 2008; Green et al., 2012). Shallow reef fish species generally have higher individual and population growth rates when compared to deep sea fish species (Rogers, 1994; Norse et al., 2012). Therefore, predation by lionfish may have greater potential for damage to native fish communities in the upper-bathyal zone. With so few records of bathyal lionfish and no quantitative estimates of lionfish densities, the ecological impacts at >200 m depth is unknown.

Current lionfish management is highly biased towards shallow reef habitats, with diver-conducted culling the major control measure implemented in the western Atlantic (Morris et al., 2009). While shallow reef culling has been found to reduce lionfish densities (Frazer et al., 2012) and help native fish populations recover (Green et al., 2014), a recent study has suggested strong depth-specific effects of culling on lionfish densities, with substantial lionfish populations remaining on MCEs despite shallow culling (Andradi-Brown et al., 2017b). Previous modelling studies have highlighted that substantial deep refuges for lionfish have the potential to undermine current management programmes (Arias-González et al., 2011). Therefore, if lionfish are widespread in the 200–300 m depth range across the western Atlantic this raises further challenges for lionfish management. There are currently few effective methods for lionfish removal in water too deep for diving, with trapping being the only widely used method. In Bermuda, lobster traps have been used to remove lionfish from MCEs, with trap modifications substantially reducing bycatch of other fish species (Pitt & Trott, 2015). Though measures of trapping effectiveness for reducing deep lionfish populations are still lacking. Traps could be trialled deeper for lionfish control in the 200–300 m range, and cameras used to monitor effects on lionfish densities.

This study documents non-native lionfish in the upper-bathyal zone in Bermuda, and Roatan, Honduras for the first time. Our observations, combined with other lionfish records, suggest that lionfish could potentially be present in 200–300 m depth habitat in many locations in the western Atlantic. Further surveys should be conducted to assess how widely lionfish are using upper-bathyal habitats, and to establish their population densities. Our results highlight the need to consider deeper lionfish populations in management programmes.

Supplemental Information

Supplemental Information 1 Lionfish GPS locations

Click here for additional data file.

Supplemental Information 2 R script to assign a depth to each lionfish record

Click here for additional data file.

We wish to thank P Stefanoudis, G Rowlands and D Gregoire-Lucente for help with this study. Nekton would like to thank SR Smith, J Pitt, T Trotts and C Flook from the Bermudian Government, and G Goodbody-Gringley from the Bermuda Institute of Ocean Sciences, for their assistance, advice and participation in the XL-Catlin Deep-Ocean Survey Bermuda Mission. We would also like to thank the crew and technicians of the Baseline Explorer, Brownies Global Logistics and Triton Submersibles. This is Nekton contribution No 1. Any use of trade, product or firm names is for descriptive purposes only and does not imply endorsement by the US Government.

Additional Information and Declarations

Competing Interests

Author Contributions

Animal Ethics

Data Availability

KS is the owner of the Roatan Institute of Deepsea Exploration. DAAB is an employee of Operation Wallacea. EG, LW and ADR are employees of the Nekton Foundation.

Erika Gress conceived and designed the experiments, performed the experiments, analyzed the data, wrote the paper, reviewed drafts of the paper.

Dominic A. Andradi-Brown conceived and designed the experiments, performed the experiments, analyzed the data, wrote the paper, prepared figures and/or tables, reviewed drafts of the paper.

Lucy Woodall and Alex D. Rogers conceived and designed the experiments, performed the experiments, reviewed drafts of the paper.

Pamela J. Schofield performed the experiments, analyzed the data, reviewed drafts of the paper.

Karl Stanley performed the experiments, reviewed drafts of the paper.

The following information was supplied relating to ethical approvals (i.e., approving body and any reference numbers):

Research permits were issued by the Department of Environment and Natural Resources, Bermuda (No. 2016070751).

The following information was supplied regarding data availability:

All raw data required for this study is contained in the Supplemental Files.

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
