# Peer review of "Lionfish (Pterois spp.) invade the upper-bathyal zone in the western Atlantic"

_PeerJ, doi:10.7717/peerj.3683_

## Round 0.1 · original submission · Minor Revisions

I have heard back from two reviewers, both of whom offered numerous comments on your manuscript. None of the comments warrant an extensive reworking of your work, and thus my decision is "minor revisions" are needed.

·

Basic reporting

no comment

Experimental design

no comment

Validity of the findings

My main critique is the statement that lionfish “may widely be using 200-300 m depth habitat in the western Atlantic”. I believe they mean widely in a geographic sense, as in lionfish are likely residing at these depths across the invaded range, but the structure of the sentence lends to the idea that lionfish are pervasive at these depths. Additionally, the term “using” is misleading. Using the habitat for what? The data provided merely show presence in the habitat, thus habitat usage cannot be speculated. I would suggest that this statement be significantly restructured.

Additional comments

This manuscript is quite simple, describing observations of invasive lionfish in deep sea regions of Bermuda and Honduras. The authors use manned submersibles in each location and visual sightings were recorded. In addition, they provide reference to another deep sea sighting in the Bahamas. While there is no experimental component to the work and no real statistical analyses involved, except for data mining of the USGS records, the results are still relevant and will be useful in our understanding of the lionfish invasion in the Atlantic. However, as the authors note, this is not the first report of lionfish below 200m, but substantiates the previous report. I have very few critiques as the manuscript is well written and the results are clearly explained.

Minor points:
1. British vs. American spelling of “metres” vs. “meters”, unclear which is preferred for PeerJ. (Line 116)
2. Line 127 change “well know/established” to “well known/established”
3. Line 258 change “Bermudan” to “Bermudian”

Figure 2. Are both images from 297m? If so, is there no image to record the presence of the fish at 304m?

Reviewer 2 ·

Basic reporting

I had to point out a lot of locations where commas were missing. Manuscript should be reviewed for punctuation.

Literature cited was nearly sufficient, but see attached. Also please add the depth range of Pteriois volitans and miles in the indopac.

There were no indications that the new observations reported here will be added to a database. Please add to the manuscript.

Results were relevant hypotheses.

Experimental design

This research is within the aims and scope of the journal.

Research question was well defined and relevant, and quadruples the number of locations where lionfish are known to inhabit the bathyl zone

Investigation was rigorous

Methods were generally described in sufficient detail and replicable, but the amount of effort expended on location lionfish in the Bahamas and Roatan is not stated. also, insufficient supplemental data was provided to run the R script. see attached for specific suggestions

Validity of the findings

Findings are valid, but the authors should explicitly define their use of the word "widely" because it could be misconstrued to mean extensively.

Data is adequate

Conclusions match the results (see attached for suggestion on the last paragraph)

Additional comments

2: sp refers to one species. spp refers to multiple. Change to spp because both volitans and
miles are known in the w Atlantic

Abstract:

28: under-sampled

Introduction:

46: for identification of 2 species of invasive lionfish cite Hamner, R. M., Freshwater, D. W. and
Whitfield, P. E. (2007), Mitochondrial cytochrome b analysis reveals two invasive lionfish
species with strong founder effects in the western Atlantic. Journal of Fish Biology, 71: 214–
222. doi:10.1111/j.1095-8649.2007.01575.x

Methods:

80-92: it is not clear how much effort was involved in looking for the lionfish. How many dives,
hours, or equivalent?

86: abbreviations at beginning of sentences must be spelled out. Fix throughout

Fig 1 A description: sighting point supposed to be >200m is on the 50m depth contour. Perhaps
include a note that this is a steep drop off.

99-109: see comment for lines 80-92, how much effort ?

110: How are the records in the US Geological Survey Nonindigenous Aquatic Species database
collected? Would you expect them to include deeper records or are they mostly from projects
occurring in shallower areas? If you expect deeper areas to be represented, but there are no
lionfish sightings in the deeper areas, then perhaps they are not widely found at depths
>200m?



118, this makes it sound like you only use the records reported in meters. Did you convert other
units to meters?



130 Include Lionfish GPS Locations from USGS database.csv as ESM1. This will make the R code
in ESM2 useable. ESM2 change heading to match function of script from line 132. It does not
“identify lionfish from >200 m depth”, it “assigns a depth to each lionfish record”.


Results:

137: be specific on how many lionfish were observed. Is it two?

Figure 2. Clarify figure caption. Are both of these photos from the same depth and location? Or
is one of them the individual seen at 304m? 2 lionfish observed on 1 day >200m in Bermuda;
any evidence that this is a regular occurrence?

Fig 3A. 180m is not >200m

146: “the Idabel has regularly observed lionfish to a maximum depth of 250 m.” Do they keep
records of these? Number of lionfish, number of days seen at depths >200m?

159: insert comma after “descriptions”

162: how often? Be explicit

162-4: rephrase to “…detail to confirm or reject it as a sighting from >200m”

170: add reference for previously reported observation

Discussion:

177: upper bathyl zone

179: northern limit of overwintering range, but juveniles make it at least up to NY

180: change to “…deepest we surveyed in this area.”

178-180: results rehash, remove

184-185: lionfish are using deep-sea habitats. The use may be more widespread and common
than presently understood.

185-: undersampled

186: insert comma after “database”

186-189: results rehash, remove

189: replace “these” with what the pronoun refers to

194: insert comma after “reason”

195: remove “when scrutinized, “ it makes the sentence awkward with all the commas

196: remove “able to be”

200: insert comma after “therefore” and “bathymetry”

200:-202: awk, try to restate

203: observations

204: replace the first “and” with a comma

205-206: this info on number of dives should be in the methods

207: insert comma after conditions

208: insert comma after light

213: insert comma after “Therefore”, fix throughout

215: delete “so”

217: delete first “lionfish”

218: new paragraph

219: insert semicolon before “therefore”, comma after

228-230: this was stated in the intro. No need to restate. Are there examples of algal-
dominated communities in the bathyl zone of the Caribbean or Gulf? I think there are in
Hawaii, maybe.

250: how many other species are caught in the traps? Is this a good suggestion, or could they
be just as bad for native species as lionfish predation?

252-3: “…suggest that surveys should be conducted to see if lionfish are widely using…”

251-254: Be specific that widely means over a substantial distance and no abundance. Also,
perhaps this is a good place to call for more surveys of lionfish in the bathyl zone because we
don’t know whether it is being utilized extensively by lionfish. Perhaps there should be extra
effort to remove lionfish with haplotypes favorable for exploiting deeper habitats to prevent a
mass expansion into the bathyl. Or perhaps lionfish will never extensively utilize these areas.
More research is required.

---

## Round 0.2 · accepted · Accept

The revision has been well done, and the manuscript is now ready to be published.

Please note two small comments from myself that should be addressed at the proof stage if not earlier:
1. The "spp." in the title should NOT be in italics.
2. Please add a 50 m benthic contour to figure 1b (Roatan) to match that in 1a.

I look forward to seeing the published version of your manuscript.